# Molecular Targeting of the Isocitrate Dehydrogenase Pathway and the Implications for Cancer Therapy

**DOI:** 10.3390/ijms25137337

**Published:** 2024-07-04

**Authors:** Stanislav Ivanov, Olger Nano, Caroline Hana, Amalia Bonano-Rios, Atif Hussein

**Affiliations:** Memorial Cancer Institute, Memorial Healthcare System, Pembroke Pines, FL 33028, USA; onano@mhs.net (O.N.); abonanorios@mhs.net (A.B.-R.); ahussein@mhs.net (A.H.)

**Keywords:** IDH, isocitrate dehydrogenase, AML, MDS, glioma, cholangiocarcinoma, ivosidenib, enasidenib, vorasidenib, olutasidenib, oncometabolite

## Abstract

The advent of comprehensive genomic profiling using next-generation sequencing (NGS) has unveiled an abundance of potentially actionable genetic aberrations that have shaped our understanding of the cancer biology landscape. Isocitrate dehydrogenase (IDH) is an enzyme present in the cytosol (IDH1) and mitochondria (IDH2 and IDH3). In the mitochondrion, it catalyzes the irreversible oxidative decarboxylation of isocitrate, yielding the production of α-ketoglutarate and nicotinamide adenine dinucleotide phosphate (NADPH) as well as carbon dioxide (CO_2_). In the cytosol, IDH catalyzes the decarboxylation of isocitrate to α-ketoglutarate as well as the reverse reductive carboxylation of α-ketoglutarate to isocitrate. These rate-limiting steps in the tricarboxylic acid cycle, as well as the cytoplasmic response to oxidative stress, play key roles in gene regulation, cell differentiation, and tissue homeostasis. Mutations in the genes encoding IDH1 and IDH2 and, less commonly, IDH3 have been found in a variety of cancers, most commonly glioma, acute myeloid leukemia (AML), chondrosarcoma, and intrahepatic cholangiocarcinoma. In this paper, we intend to elucidate the theorized pathophysiology behind IDH isomer mutation, its implication in cancer manifestation, and discuss some of the available clinical data regarding the use of novel IDH inhibitors and their role in therapy.

## 1. Introduction

The fundamental abnormality resulting in the development of cancer is the continuous unregulated proliferation of clonal cells. This unregulated proliferation is a result of the net accumulation of anomalies in multiple cell regulatory systems. Such regulatory systems include checkpoint inhibition and triggered cell death, DNA repair mechanisms, and resilience to epigenetic stressors and oncometabolites. Cells that have acquired the ability to bypass these and other regulatory systems result in malignant proliferation that is able to disrupt normal tissue and spread to distant parts of the body, contributing to end-organ damage. Recent developments in genomic profiling via next-generation sequencing (NGS) have identified a multitude of targetable mutations that are presently being researched as potential avenues for therapy. One such target is the isocitrate dehydrogenase (IDH) enzyme system. This protein product, found both in the cytosol and mitochondria of cells, plays a key role in cellular metabolism, growth, and differentiation. Mutations in the catalytic domain of IDH can result in fundamental impairments of the cell cycle but may also result in synthesis of oncometabolites that can promote aberrant cellular growth and thus lead to the development of malignancy. Additionally, elucidating the intricate interplay between IDH mutations and other molecular pathways involved in carcinogenesis is crucial for the development of targeted therapies aimed at halting tumor progression and improving patient outcomes.

## 2. Biology of Isocitrate Dehydrogenase

Isocitrate dehydrogenase is a family of proteins that can be classified as either homodimeric or heterotetrameric in nature, contingent on their function and component subunits. The homodimeric isoforms, IDH1 and IDH2, are composed of two identical subunits, while IDH3 is a heterotetramer. Five genes encode for the IDH family of genes. IDH1, encoded by the *IDH1* gene on 2q.33.3, is found in the cytosol and catalyzes the oxidative decarboxylation of isocitrate (ICT) to α-ketoglutarate (2KG), as well as the reverse reductive carboxylation of 2KG to ICT. IDH2, encoded by the *IDH2* gene on 15q26.1, catalyzes the same reactions as IDH1, except it does so in the mitochondria. The functions of these two enzymes play an important role in lipogenesis and glycolysis regulation through ICT synthesis in glutamine metabolism and thus exert selective pressure in cell propagation and differentiation [1,2,3,4].

Unlike IDH1 and IDH2, IDH3 is encoded by several genes, one for each of its subunits: *IDH3A* on 15q25.1-q25.2, *IDH3B* on 20p13, and IDH3G on Xq28. This isoform of IDH is primarily responsible for the irreversible conversion of ICT to 2KG as part of the tricarboxylic acid. The function of the IDH isomers is markedly dependent on the energy state of the cells. In times of low energy states, metabolites can be processed into energy by the production of reduced electron carriers like 1,4-dihydronicotinamide adenine dinucleotide (NADH), which, in turn, are utilized by the mitochondrial electron transport chain to produce adenosine triphosphate (ATP). Conversely, in states of high energy (high NADH/NAD ratio), mitochondrial isocitrate stores are shunted to the cytoplasm, where IDH1 and IDH2 play active roles in the production of nicotinamide adenine dinucleotide phosphate (NADPH), which is used to reduce glutathione and participate in the defense against DNA damage from reactive oxygen species as well as repair of mitochondrial oxidative damage. Alterations in the function of these enzyme catalysts can lead to dysregulation in cell differentiation and homeostasis [1,2,3,4] (See Figure 1A).

## 3. Pathophysiology of IDH Mutants

Understanding the physiologic role of the IDH isomers in normal cells helps elucidate the mechanisms in which these isomers contribute to malignant cellular proliferation. Interestingly, in tumors where the anabolic state is maximized, and the cells have reached high intramitochondrial NADH/NAD ratio, the tricarboxylic acid cycle (TCA) is arrested, while the activity of IDH1 and IDH2 remains elevated. This pattern of activity suggests two important concepts: (1) accumulated intramitochondrial citrate metabolites are shunted to the cytoplasm where they are used in an anabolic cascade to produce fatty acids and phosphoglycerides needed for synthesis of biologic membranes; and (2) mutations in IDH3 are not contributory to pro-oncogenic progression of cells as much as mutations in IDH1 and IDH2.

The first instance of a cancer associated IDH mutation was described in a patient with colorectal cancer. Later, with the advent of whole-genome sequencing, mutations in IDH 1 and IDH2 were noted in patients with glioblastoma multiforme, as well as a few hematopoietic neoplasms including myeloid neoplasms, angioimmunoblastic T-cell lymphoma, lung cancer, chondrosarcomas, and cholangiocarcinomas. The majority of IDH1 and IDH2 mutations have been attributed to an arginine residue within the catalytic pocket of the enzyme. Mutations in the IDH1 occur position 132, with the most common variants being R132H, R132C, R132L, R132S, and R132G.

Of these variants, R132H is seen in most all pathology types; however, it does have higher frequency of expression in patients with newly diagnosed gliomas, approximately 85% of the reported IDH1 mutants. This characteristic is important, as it is key in the diagnostic workup needed to correctly identify this population of patients. In certain instances, for example, the use of IHC (immunohistochemistry), which can detect R132H, may not be sufficient as it does have a false positive rate of approximately 5–10% and cannot accurately detect other mutational changes. As such, the use of molecular studies would be necessary to definitively rule out or rule in IDH1 mutants. With respect to IDH2, mutations commonly occur in positions 172 and 140.

The sheer clustering of the mutations within the metabolically active pocket of the enzyme suggests that these mutations are the result of a novel configuration which grants the enzyme oncogenic activity. This hypothesis was further confirmed when cells expressing mutant IDH1^R132H^ were noted to be functional but produced accumulating amounts of an enantiomer of 2KG, namely, D-2 hydroxyglutarate (D-2-HG). This is the same substance that has been noted to accumulate in increasing amounts in patients with confirmed diagnosis of gliomas and hematologic malignancies such as acute myeloid leukemia (AML). Expression of D-2-HG in confirmed cases of malignancy suggests that the IDH mutations have oncogenic potential. Moreover, this metabolite has been noted to originate from the reverse reductive carboxylation of 2KG. While important to normal cells, the reverse carboxylation of 2KG can lead to overaccumulation of D-2-HG. Mutant isoforms of IDH that favor the production of D-2-HG can therefore amplify the carcinogenic effect of this oncometabolite. It is, therefore, no surprise that cells like astrocytes and myeloid cells that normally produce high amounts of citrate can behave aggressively when mutant isoforms of IDH enzyme are present [5,6,7,8].

The metabolic product of the mutant IDH also depends on the type of mutation present. Studies have demonstrated that homozygotic mutations did not produce viable functionality, partly because of the dependence of the arginine residues in the active site and were therefore selected against. Thus, heterozygote mutants are the observed malignant variants. The additive advantage of these heterozygote mutants is that the oncometabolite, D-2-HG, not only has higher affinity for enzymatic catalyzation but also serves to suppress production of native 2KG from wild-type homodimers [9].

Nonetheless, such a discovery was not sufficient to determine whether IDH mutations alone account for the development of malignancy. Previous beliefs that IDH acts as a tumor suppressor were negated by in vitro transformation assays which noted that it was the neomorphic IDH1 and 2 isoforms that promoted the proliferation and inhibited the differentiation of immortalized human cells. These mutations alone, however, were not sufficient to produce malignancy, and it is therefore believed that IDH mutations should occur in the context of other carcinogenic mutations to manifest as a truly malignant process (see Figure 1B).

## 4. Myeloid Neoplasms—Acute Myeloid Leukemia/Myelodysplastic Syndrome

Although initially described in colon cancer and glioma patients, IDH mutations were soon described in myeloid malignancies, particularly myelodysplastic syndrome (MDS) and AML. The most common mutations noted in IDH1 were seen with the R132C and R132H variants, whereas the mutations noted with IDH2 were R140Q and R172K. As with the previously noted in vitro studies of isocitrate dehydrogenase, mutants were noted to produce high levels of D-2-HG. Thus, the driver leukemia is in part through D-2-HG inhibition of TET2, an α-ketoglutarate-dependent enzyme involved in demethylation and regulation of epigenetic status [10]. As D-2-HG accumulates, it severely attenuates TET2-dependent demethylation of genomic DNA, which contributes to the pathogenesis of acute myeloid leukemia via metabolic and epigenetic dysregulation, including the blocking of normal hematopoietic cellular differentiation [11,12].

The gene mutations associated with the pathogenesis of myeloid diseases are missense mutations which alter the active site and grant it greater affinity for the oncometabolite which exerts its effects on cells as described previously. These mutations are seen in almost all subtypes of acute myeloid leukemia. Most patients noted to harbor these genetic aberrations tend to be older and tend to have diploid or intermediate cytogenetics. While the impact of IDH mutations on acute myeloid leukemia prognosis remains somewhat controversial, generally inferior outcomes are observed, namely, lower overall survival along with a tendency to manifest chemotherapy resistance. Nonetheless, as demonstrated in previous studies, expression of mutant IDH1 or IDH2 can be reversed by inhibitor treatment, which offers potential for treatment [11,12].

Similar findings can be seen in patients with MDS, of whom IDH1/IDH2 mutants make up approximately 5% of the cases. In this population, an isocitrate dehydrogenase enzyme mutation also affects the MDS phenotype. IDH-mutated myelodysplastic syndrome is associated with a lower absolute neutrophil count, higher platelet count, and higher bone marrow blast percentage at diagnosis, and it occurs most often in the setting of diploid karyotype (60%), trisomy 8 (10%), or other intermediate-risk (23%) cytogenetic abnormalities [11,12,13,14].

### 4.1. Enasidenib

The first-of-its-kind oral selective IDH2 inhibitor, specifically targeting mutant IDH2 in patients with AML/MDS, enasidenib, received approval from the FDA in 2017. The medication is a selective allosteric inhibitor of IDH2 which stabilizes the mutated IDH2 enzyme, thus inhibiting the conversion of 2-KG to D-2-HG [13]. The reduction in the oncometabolite results in the release of the differentiation block allowing maturation into normal functional cells. The approval was granted on a phase 1/2 dose-escalation trial of 345 patients. The patients were treated with a maximum daily dose of 100 mg, leading to marked reduction in D-2-HG production. The overall response rate (ORR) was 38.8%, with a median duration of 5.6 months. Complete remission (CR) with incomplete hematologic recovery and CR with incomplete neutrophil or platelet recovery (CRi/CRp) was achieved in 29.0% of patients (CR 19.6%, CRi/CRp 9.3%), and the median overall survival (OS) among relapsed/refractory patients was 8.8 months. The most common side effects included indirect hyperbilirubinemia and thrombocytopenia, as well as IDH differentiation syndrome [15]. 

Patients treated with enasidenib demonstrated reduction in the level of oncometabolite, as was variant allele expression; however, such reduction did not correlate with clinical outcomes. Data also suggest that enasidenib promotes differentiation of leukemic blasts, thus enforcing the idea that most patients would benefit from combination therapy with chemotherapeutics to clear the blast burden.

In the first line, the phase II portion of an open-label, randomized phase I/II study (NCT02677922) compared 75 mg/m^2^/day × 7 day/cycle azacitidine alone versus a combination with enasidenib 100 mg every day. Preliminary data demonstrate that enasidenib plus azacitidine significantly improves response rates and durations of response. The data also showed that the addition of enasidenib was well tolerated in older patients with mutant IDH2 newly diagnosed AML. The ORR reported was 48% with combination therapy versus 14% with single-agent azacitidine. The median OS was 22 months in both treatment arms and longer EFS in the combination therapy arm [15]. Additional studies are looking at the use of single-agent enasidenib for maintenance therapy after induction or post-allogeneic transplant for those patients who continue to demonstrate IDH2 mutations.

The management of MDS has been a topic of much discussion. Specific first-line treatments have been outlined for patients with low-risk disease and patients with loss of the long of arm of chromosome 5 (del5q). Many such patients have been treated with erythropoietin or lenalidomide, respectively. Similarly, MDS with ringed sideroblasts or transfusion-dependent MDS not responding to erythropoietin have been managed with luspatercept. All other MDS groups have, by convention, received therapy with a hypomethylating agent (HMA) such as azacitadine or decitabine. Undeniably, most of this latter group tend to develop resistance to therapy with single-agent HMA. DiNardo et al. analyzed the safety and efficacy of the combination of azacitidine and enasidenib for treatment-naïve, higher-risk MDS and IDH2 mutation, as well as enasidenib monotherapy in MDS with IDH2 mutation after prior HMA therapy. Enasidenib therapy was, overall, well tolerated [16]. The frequency of indirect hyperbilirubinemia (14%) and IDH-related differentiation syndrome (IDH-DS) (16%) was similar to that in previous reports of enasidenib therapy, and IDH-DS was effectively managed with initiation of systemic corticosteroids and supportive care measures [16].

In the cohort of treatment-naïve mutant IDH2 MDS, the ORR was 74%, with composite CR of 70%; 37% of patients attained either full CR (*n* = 7) or mCR with hematologic improvement (*n* = 3). These results compared favorably with anticipated outcomes with azacitidine monotherapy (ORR: 35–40%) and correlated well with long-term survival of 28 months in responding patients [16].

### 4.2. Ivosidenib

Ivosidenib received approval in 2018. Like enasidenib, this medication depends on the suppression of D-2HG production, unlike enasidenib, however, it exerts its effect on the mutant IDH1 enzyme. Ivosidenib competes with magnesium ions, a key cofactor in the formation of catalytically active site, thus preventing peripheral formation of the oncometabolite. The use of ivosidenib results in increased levels of cell-surface markers of differentiation and increases in the proportion of mature myeloid cells. A phase 1/2 trial consisted of a dose-escalation phase with 258 patients, with 179 patients in the setting of R/R disease. Based on the results of the study, the recommended dose of ivosidenib was set to 500 mg daily. This dose was correlated with efficacy and reduction of serum 2-HG. CR/CRh was 30.4% (21.6% CR), ORR was 41.6%, median OS was 8.8 months, and 18-month survival of those in CR/CRh was 50.1%. Grade 3 to 4 AEs included prolongation of the QT interval (7.8%), IDH differentiation syndrome (3.9%), anemia, (2.2%), and thrombocytopenia (3.4%).

In May of 2019, ivosidenib was also approved for frontline AML in patients with comorbidities precluding the use of intensive induction chemotherapy. In this population, ivosidenib achieved a CR/CRh rate of 42.4% (CR 28.6%, CRh 14.3%) and a median OS of 12.6 months. This is an important outcome in and of itself, as it is associated with improved quality of life, particularly in rural or elderly patients with limited means of transportation to a clinic [17,18,19,20,21].

### 4.3. Olutasidenib

In 2022, yet another IDH1 inhibitor was approved for the management of relapsed or refractory AML. In an open-label, multicenter study, 153 patients with R/R AML and a median age of 71 years received ≥1 dose of olutasidenib monotherapy (phase I identified a dose of 150 mg, twice daily, in continuous 28-day cycles). Of these patients, 66% had de novo AML and 34% had secondary AML. The majority (73%) of patients had intermediate cytogenetics, with 17% and 4% having poor and favorable cytogenetics, respectively. The primary endpoint was CR plus CR with partial hematologic recovery. Of the 147 patients evaluable for response, the CR+CRh rate was 35%, with a median time to CR+CRh response of 1.9 months and median duration of CR+CRh response of 25.9 months [20].

## 5. Glioblastoma Multiforme

Glioblastoma multiforme (GBM) is an invasive brain tumor that has been notoriously refractory to chemotherapy and radiation. In 2008, Parsons et al. attempted to identify novel targets for possible therapeutic interventions [21]. The project utilized genomic sequencing strategies by looking into 22 patients with primary and secondary glioblastoma multiforme tumors. Five of the six secondary gliomas were notable for IDH mutation, whereas none of the sixteen primary gliomas were wild type. This study found that somatic mutations in IDH were more common in certain cancers as opposed to others [21]. In years that followed, additional sequencing studies noted that IDH mutations were seen in >70% of grade II and III gliomas and in >80% of secondary gliomas. As mentioned previously, many of these mutations were noted to be mutations in IDH1, with a greater percentage (85%) representing R132H alterations in the active site.

The identification of heterozygous point mutations in the genes encoding for isocitrate dehydrogenase was a breakthrough in the understanding of the possible pathophysiology of gliomas. This offered a novel target for treatment. The pathophysiologic effect of IDH in gliomas was believed to be akin to that of other malignancies in that the accumulation of the oncometabolite inhibits a series of α-ketoglutarate-dependent kinases, which, in turn, impairs the cellular mechanisms of differentiation and cell repair. This discovery became a pivotal aspect of the diagnosis of GBM, as it allowed for subclassification and prognostication. Patients with IDH mutation usually harbor better prognosis and response to therapy. To capitalize on the discovery of IDH mutations, another barrier must be overcome, namely, the blood–brain barrier. Unlike myeloid malignancies or other solid tumors, IDH inhibition is not readily possible for tumors located in the CNS [21]. Research has focused on developing exactly this type of therapies. In August of 2023, Mellinghoff et al. published a pivotal paper detailing the benefit of vorasidenib in the treatment of IDH1 or IDH2 mutant low-grade gliomas [22].

### 5.1. Vorasidenib

Vorasidenib is a the first-in-class dual IDH1/IDH2 inhibitor which is particularly well suited to penetrate the blood–brain barrier. The rationale for the use of dual IDH inhibition has been justified, as the switching from IDH1 mutant to IDH2 mutants and vice versa has been thought to be a source of resistance to therapies, particularly in myeloid malignancies such as AML.

Mellinghoff et al. enrolled 331 patients who were randomized to receive vorasidenib (168 patients) versus a placebo (163 patients). At median follow-up of 14.2 months, 226 patients (68.3%) continued to receive vorasidenib or the placebo. The median PFS was significantly improved in the vorasidenib group (27.7 months) as compared with the placebo group (11.1 months). The hazard ratio for disease progression or death was calculated as 0.39; 95% (CI: 0.27–0.56; *p* < 0.001). The time to the next intervention was significantly improved in the vorasidenib group (HR 0.26; 95% CI, 0.15–0.43; *p* < 0.001). Grade ≥ 3 AEs occurred in 22.8% of the patients who received vorasidenib versus 13.5% in the placebo group. An increased alanine aminotransferase level of grade ≥ 3 occurred in 9.6% of the patients who received vorasidenib versus none in the placebo group.

The study demonstrated that a grade 2, IDH-mutated glioma patient who received vorasidenib derived benefit in progression-free survival and also noted delayed time to the next intervention [22].

### 5.2. Cholangiocarcinoma

Biliary tract cancers (BTCs) have been long treated with gmcitabine and cisplatin, based on the ABC-02 trial which showed improved overall survival (OS) [23]. Then addition of immunotherapeutic PD-L1 inhibitor, durvalumab, to that regimen was shown to further improve the OS based on the TOPAZ-1 trial [24]. During relapsed settings, the ABC-06 trial showed that therapy with mFOLFOX (5-Flurouracil, oxaliplatin, and leucovorin) is beneficial, compared to symptom control [25]. Despite recent advances, cholangiocarcinoma (CCA) remains one of the most challenging cancers to treat, with very poor outcomes. Only 10–20% of cases are amenable to curative-intent surgery, and, even in resected cases, the 5-year overall survival is less than 50%, while the unresectable cases have a median overall survival close to a year [26]. In a retrospective analysis of patients with IDH-mutated cholangiocarcinoma, the median OS and PFS were 21.2 months and 8.3 months, respectively [26]. This highlights a great need for more advances in biliary tract carcinomas. A study of intrahepatic cholangiocarcinoma by Xiang et al. demonstrated significant intra-tumoral heterogeneity across genomic, transcriptomic, and immune levels [27]. The authors noted intra-tumoral heterogeneity in IDH-mutant subgroups, including IDH-mutated tumors and IDH-like tumors which are, in essence, IDH-wild-type but exhibit similar molecular profiles [27]. This group of cholangiocarcinomas is characterized by a cold tumor microenvironment with less T-cell infiltration and activation. Tumor profiling revealed that nearly 40% of biliary tract carcinomas harbor potentially actionable aberrations. IDH 1/2 mutations occur in nearly 10% of biliary tract carcinoma cases. It has been reported to be more common in the intrahepatic than the extrahepatic CCA [28].

In a study of whole-genome and epigenomic landscapes of CCA, IDH 1/2-mutated CCA were classified as cluster 4, and were mainly non-fluke-initiated tumors with a better overall survival compared to clusters 1 & 2 [29].

IDH 1/2 mutations result in elevated levels of the oncometabolite 2-hydroxyglutarate (2-HG). Elevation of 2-HG is associated with higher DNA CpG methylation and altered histone methylation. Isocitrate dehydrogenase mutations cause alterations in the hypoxia signaling, causing increased hypoxia inducible factor level in the tumor microenvironment with resultant angiogenesis and tumor survival. In mice models, IDH mutation with the resultant elevation in 2-HG promoted cholangiocarcinoma tumor maintenance through an immune evasion program centered on dual 2-HG-mediated mechanisms: suppression of CD8+ T-cell activity and tumor cell-autonomous inactivation of TET2 DNA demethylase [30].

Many researchers have investigated targeting IDH mutations for treatment of cancer, including AML and solid tumors (especially gliomas). In the phase 1 study of IDH-1 inhibitor Ivosedinib in CCA, gliomas, and chondrosarcomas, the drug showed decreased levels of 2-HG by up to 98%. Ivosedinib was then examined in the phase III ClarIDHy trial in patients with previously treated IDH-1 mutant CCA, who had up to two prior lines of therapy. Two hundred thirty patients were randomly assigned to receive ivosedinib 500 mg daily or placebo. The median PFS was 2.7 months in the ivosedinib compared to 1.4 months in the placebo group (*p* < 0.0001). Of the 124 patients in the ivosidenib group, 63 (51%) had stable disease compared to 17 (28%) of 62 patients in the placebo group [31,32,33].

Several other selective and non-selective IDH-1 mutation inhibitors are under investigation in phase I/II trials. For example, olutasidenib was investigated in a phase Ib/II trial for solid tumors at a dose of 150 mg twice daily, orally. The trial included 44 patients with relapsed or refractory IDH-1 mutant solid tumors, of which, 26 patients had intrahepatic CCA. This preclinical study demonstrated the safety and tolerability of olutasidenib in relapsed and refractory IDH-1-mutated solid tumors [34].

## 6. Future Direction

The biologic effects of IDH mutations, such as the overproduction of D-2-HG, are common and independent of the cancer type; however, major differences exist in the way patients respond to therapy, suggesting that there are tissue-dependent and even differentiation-stage-dependent phenotypes that affect responses to therapy. As such, trials are on the way to understand novel approaches that can be used to for the treatment of IDH mutant tumors, both liquid and solid. Table 1 lists some of the ongoing trials in the field.

Additional work is being done to investigate the use of IDH inhibitors in conjunction with DNMT inhibitors, as both types of agents have previously demonstrated effect on IDH mutant gliomas.

Moreover, therapies beyond small-molecule inhibitors are being looked for, harboring potential for novel therapies such as neoepitope vaccines. One such epitope was noted in gliomas harboring IDH1^R132H^. This epitope can be seen presented to major histocompatibility complex (MHC) class II, resulting in the formation of CD4+ T-helper/T-cell response that is effective in controlling tumor growth. The believed mechanism of tumor growth control has been thought to be secondary to the release of pro-inflammatory cytokines such as interferon-γ and TNF-α.

In addition to combinations with IDH inhibitors, vaccine therapies are also being investigated in combination with immune checkpoint inhibitors such as avelumab in an attempt to augment the vaccine-induced T-cell responses. The rationale behind these approaches is to blunt the immune evasion of malignant cells that are otherwise able to counteract the function of PD1-expressing cells in the tumor microenvironment. The safety and efficacy of such combinations is under investigation. The ongoing NOA21 trial randomized the phase 1 window-of-opportunity design, the safety and immunogenicity of IDH1-vac in combination with the programmed death-ligand 1 (PD-L1)-blocking immune checkpoint inhibitor avelumab (NCT03893903) [35,36,37].

## 7. Conclusions

Considerable progress has been made in the understanding of IDH-mutant malignancies, particularly in elucidating their pathogenesis, mutational patterns, and response to therapy. However, substantial further research is needed to effectively translate our understanding of the biological nuances of these mutations into clinical practice. It is worth noting that mutation of the IDH protein may not be the sole factor leading to the development of malignancy; other genetic and environmental factors could play significant roles.

Much remains to be determined regarding the intricate landscape of cell metabolism that may contribute to the development of IDH mutations and their oncogenic sequelae. Currently, treatment of malignant processes with IDH mutations amenable to targeted therapy is predominantly reserved for diseases that have relapsed following initial therapy or proven refractory to first-line therapies. Formulating clinical trials is imperative to ascertain whether IDH inhibition in the frontline setting would yield augmented clinical benefits for patients. It is also essential to assess if patients with these disease states would experience enhanced overall survival and/or quality of life. Additional investigations are crucial to fully elucidate the broad spectrum of implications these mutations pose in clinical settings and to optimize therapeutic strategies accordingly.

## Figures and Tables

**Figure 1 ijms-25-07337-f001:**
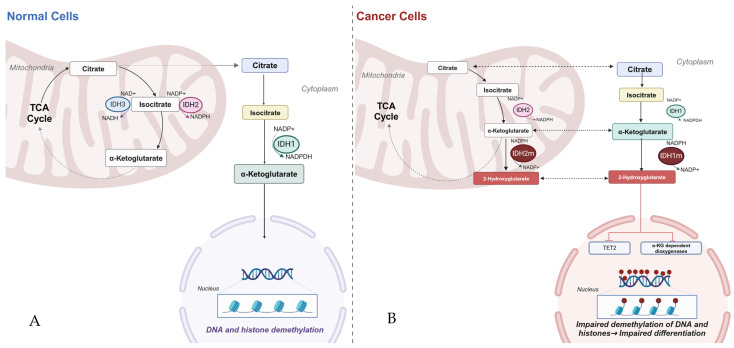
(**A**) Biologic function of IDH isomers. (**B**) Pathogenic function of mutant IDH isomers in malignant cells.

**Table 1 ijms-25-07337-t001:** Current phase I/II trials exploring IDH inhibitors in treatment of solid tumors.

Clinical Trial	Compound	Mechanism	Disease	Trial Phase
NCT04762602	HMPL-306	Dual IDH 1/2 inhibitor	Solid tumors	I
NCT02381886	IDH 305	Inhibition of IDH1 with R132 Mutations	Advanced malignancies that harbor IDH1R132 mutations	I
NCT02273739	Enasidenib (AG-221)	IDH2 inhibitor	Advanced solid tumors with IDH2 mutation	I/II
NCT02193347	PEPIDH1M vaccine	Vaccine	Gliomas, WHO grade 2, mIDH	I
NCT 03893903	IDH-Vac + Avelumab	Vaccine	Glioma, mIDH	I
NCT02771301	IDH1R132H-DC vaccine	Vaccine	Glioma, mIDH	I

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
