# Peer review of "Molecular Targeting of the Isocitrate Dehydrogenase Pathway and the Implications for Cancer Therapy"

_ijms, 2024, doi:10.3390/ijms25137337_

Round 1

Reviewer 1 Report

Comments and Suggestions for Authors

The review article by Ivanov et al. discusses the role of IDH mutations in carcinogenesis of specific tumor types and describes recent therapeutic approaches for improving outcomes in patients with IDH-driven tumors. Overall, this article was informative, easy to read and summarized the IDH enzyme and pathway(s) well. I do have a few minor comments for consideration to further improve the review article:

1-The figure provided is too small and difficult to read in the draft. Please increase size.

2- I suggest adding a line or two stating clearly that R132H alterations are the most common in brain tumors whereas non-R132H mutations are more common in the other described tumors (would include percentages of R132H vs non R132H for different tumor types). This is important for understand the role of targeted therapies as well as understanding the differences for diagnostics (that drive therapy selection). For example, a R132H IHC is routinely used to detect this variant in brain tumors clinically, yet false negatives can occur because 5-10% of IDH mutations are not R132H and it's recommended that molecular testing be performed if IHC is normal. It's also not uncommon that people use IHC to assess for IDH mutations in other malignancies not understanding that most non-brain tumors rarely have the R132H mutation (what IHC targets), which again results in a false negative finding. Therefore, knowing differences in specific mutations for the different tumor types is relevant.

3-In the conclusion of the review article, the authors correctly state that IDH mutations are almost always part of a more complex genetic and environmental story. Because other genetic alterations occur in other pathways in patients with IDH mutations, it may be beneficial for the authors to briefly mention studies assessing combination therapies (IDH targeted drugs with drugs targeting other commonly altered pathways/immunotherapy approaches).

Author Response

1-The figure provided is too small and difficult to read in the draft. Please increase size.

Image was resized to the best of ability to improve visibility.

2- I suggest adding a line or two stating clearly that R132H alterations are the most common in brain tumors whereas non-R132H mutations are more common in the other described tumors (would include percentages of R132H vs non R132H for different tumor types). This is important for understand the role of targeted therapies as well as understanding the differences for diagnostics (that drive therapy selection). For example, a R132H IHC is routinely used to detect this variant in brain tumors clinically, yet false negatives can occur because 5-10% of IDH mutations are not R132H and it's recommended that molecular testing be performed if IHC is normal. It's also not uncommon that people use IHC to assess for IDH mutations in other malignancies not understanding that most non-brain tumors rarely have the R132H mutation (what IHC targets), which again results in a false negative finding. Therefore, knowing differences in specific mutations for the different tumor types is relevant.

Commentary detailing the above specifics of R132H mutation has been included in two sections: “Pathophysiology of IDH mutants” Line 106 as well as “ Glioblastoma Multiforme” Line 294.

3-In the conclusion of the review article, the authors correctly state that IDH mutations are almost always part of a more complex genetic and environmental story. Because other genetic alterations occur in other pathways in patients with IDH mutations, it may be beneficial for the authors to briefly mention studies assessing combination therapies (IDH targeted drugs with drugs targeting other commonly altered pathways/immunotherapy approaches).

Could not locate studies with reliable data on ongoing combination therapies aside from those included in Table 1.

Reviewer 2 Report

Comments and Suggestions for Authors

In this review paper, the authors conduct an analysis of IDH enzyme involvement in cancer biology and try to elucidate the pathophysiology behind the IDH isomer mutations. It is a comprehensive review paper and no self-citations were detected. Although it is an interesting approach and the paper structure allows an overview of the IDH involvement in different malignant pathologies, underlying the (potential) implications in cancer therapy, the paper has few aspects that need to be improved:

1.     There is no affiliation of the authors Olger Nano, Caroline Hana and Amalia Bonano-Rios, Atif Hussein. Please write the affiliation for these authors.

2.     Line 85: “T -cell lymphomas” – please write it as T-cell lymphoma

3.     Paragraph starting at line 112 has no reference, although it states that “Previous beliefs that IDH acts as a tumor suppressor…”. Please add reference

4.     Figure 1 – Is it an original figure or is adapted, or taken from some other paper? If it is not original, please add reference.

5.     Line 175 – “75 mg/m2/day x 7 day/cycle”. Please write as 75 mg/m2/day…

6.     Line 181 – “Additional studies are looking ….”. There is no reference provided for this paragraph. Please include the references of these studies you mention.

7.     Line 185 – Please rephrase the paragraph because it is very hard to understand: “The management of MDS has been a marked topic of discussion with specific first line treatment outlined for patient with low-risk disease and 5q- in the role of lenalidomide as 186 well as patients with MDS with ringed sideroblasts or transfusion dependent MDS with not responding to erythropoietin, in the form of luspatercept.” Please explain what is “5q”.

8.     Line 189 – “agent hypomethylating agent (HMA)”. Please remove one of the words “agent”.

9.     Line 191 – “DiNardo et al….”. Please add the reference number for this author.

10.   Line 238 – “In 2008 Parson et al….”. Please add the reference number for this author.

11.   Line 257 – “In August of 2023 Mellinghoff et al…”. Please add reference number for this author.

12.   Line 257 – “In August of 2023 Mellinghoff et al. published a groundbreaking paper in NEJM detailing the benefit of Vorasenib in the treatment of IDH1 or IDH2 258 mutant low-grade gliomas.” Please rephrase.

13.   Line 294 – “…Xiang et al….”. Please add reference number for this author.

14.   Table 1. Please include all the current phase I/II trials, not just few examples. 

15.   Reference section – References are written in different styles. Please keep a unitary style for reference formatting.  

Author Response

  1. There is no affiliation of the authors Olger Nano, Caroline Hana and Amalia Bonano-Rios, Atif Hussein. Please write the affiliation for these authors.

            Affiliations are provided for all authors.

  1. Line 85: “T -cell lymphomas” – please write it as T-cell lymphoma

                       Corrected as advised.

  1. Paragraph starting at line 112 has no reference, although it states that “Previous beliefs that IDH acts as a tumor suppressor…”. Please add reference.

            Reference Added

  1. Figure 1 – Is it an original figure or is adapted, or taken from some other paper? If it is not original, please add reference.

            Figure 1 is an original product generated via BioSketch software. No need to reference.

  1. Line 175 – “75 mg/m2/day x 7 day/cycle”. Please write as 75 mg/m2/day…

            Corrected as advised.

  1. Line 181 – “Additional studies are looking ….”. There is no reference provided for this paragraph. Please include the references of these studies you mention.

            References added.

  1. Line 185 – Please rephrase the paragraph because it is very hard to understand: “The management of MDS has been a marked topic of discussion with specific first line treatment outlined for patient with low-risk disease and 5q- in the role of lenalidomide as 186 well as patients with MDS with ringed sideroblasts or transfusion dependent MDS with not responding to erythropoietin, in the form of luspatercept.” Please explain what is “5q”.

            Corrected as suggested. Extrapolated on 5q and defined it as the short arm of chromosome 5.

 Line 189 – “agent hypomethylating agent (HMA)”. Please remove one of the words “agent”.

Revised as advised

  1. Line 191 – “DiNardo et al….”. Please add the reference number for this author.

Reference added.

 Line 238 – “In 2008 Parson et al….”. Please add the reference number for this author.

Reference added.

  1. Line 257 – “In August of 2023 Mellinghoff et al…”. Please add reference number for this author.

Reference added.

  1. Line 257 – “In August of 2023 Mellinghoff et al. published a groundbreaking paper in NEJM detailing the benefit of Vorasenib in the treatment of IDH1 or IDH2 258 mutant low-grade gliomas.” Rephrase.

    Rephrasing removed “use of NEJM”. Rephrasing as: “ In August of 2023 Mellinghoff et al. published a pivotal paper detailing the benefit of Vorasenib in the treatment of IDH1 or IDH2 mutant low-grade gliomas.”

  2. Line 294 – “…Xiang et al….”. Please add reference number for this author.

            Reference added

 Table 1. Please include all the current phase I/II trials, not just few examples.

These are all the relevant clinical trials listed in Table 1  

  1. Reference section – References are written in different styles. Please keep a unitary style for reference formatting.  

       Reference section formatted to the same font.